# Screening of Volatile Compounds, Traditional and Modern Phytotherapy Approaches of Selected Non-Aromatic Medicinal Plants (*Lamiaceae, Lamioideae*) from Rtanj Mountain, Eastern Serbia

**DOI:** 10.3390/molecules28124611

**Published:** 2023-06-07

**Authors:** Milica Aćimović, Jovana Stanković Jeremić, Ana Miljković, Milica Rat, Biljana Lončar

**Affiliations:** 1Institute of Field and Vegetable Crops Novi Sad—IFVCNS, National Institute of the Republic of Serbia, Maksima Gorkog 30, 21000 Novi Sad, Serbia; 2Institute of Chemistry, Technology and Metallurgy—ICTM, National Institute of the Republic of Serbia, University of Belgrade, Njegoševa 12, 11000 Belgrade, Serbia; jovanas@chem.bg.ac.rs; 3Faculty of Medicine, University of Novi Sad, Hajduk Veljkova 3, 21000 Novi Sad, Serbia; ana.miljkovic@mf.uns.ac.rs; 4Faculty of Science, University of Novi Sad, Trg Dositeja Obradovića 3, 21000 Novi Sad, Serbia; milica.rat@dbe.uns.ac.rs; 5Faculty of Technology, University of Novi Sad, Bulevar Cara Lazara 1, 21000 Novi Sad, Serbia; cbiljana@uns.ac.rs

**Keywords:** *Sideritis montana*, *Teucrium montanum*, *Teucrium chamaedrys*, *Marrubium peregrinum*

## Abstract

Ironwort (*Sideritis montana* L.), mountain germander (*Teucrium montanum* L.), wall germander (*Teucrium chamaedrys* L.), and horehound (*Marrubium peregrinum* L.) are species widely distributed across Europe and are also found in North Africa and West Asia. Because of their wide distribution they express significant chemical diversity. For generations, these plants have been used as medical herbs for treating different aliments. The aim of this paper is to analyze volatile compounds of four selected species that belong to the subfamily Lamioideae, family Lamiaceae, and inspect scientifically proven biological activities and potential uses in modern phytotherapy in relation to traditional medicine. Therefore, in this research, we analyze the volatile compounds from this plants, obtained in laboratory by a Clevenger-type apparatus, followed by liquid–liquid extraction with hexane as the solvent. The identification of volatile compounds is conducted by GC-FID and GC-MS. Although these plants are poor in essential oil, the most abundant class of volatile components are mainly sesquiterpenes: germacrene D (22.6%) in ironwort, 7-*epi*-*trans*-sesquisabinene hydrate (15.8%) in mountain germander, germacrene D (31.8%) and *trans*-caryophyllene (19.7%) in wall germander, and *trans*-caryophyllene (32.4%) and *trans*-thujone (25.1%) in horehound. Furthermore, many studies show that, in addition to the essential oil, these plants contain phenols, flavonoids, diterpenes and diterpenoids, iridoids and their glycosides, coumarins, terpenes, and sterols, among other active compounds, which affect biological activities. The other goal of this study is to review the literature that describes the traditional use of these plants in folk medicine in regions where they grow spontaneously and compare them with scientifically confirmed activities. Therefore, a bibliographic search is conducted on Science Direct, PubMed, and Google Scholar to gather information related to the topic and recommend potential applications in modern phytotherapy. In conclusion, we can say that selected plants could be used as natural agents for promoting health, as a source of raw material in the food industry, and as supplements, as well as in the pharmaceutical industry for developing plant-based remedies for prevention and treatment of many diseases, especially cancer.

## 1. Introduction

Every culture has its own heritage passed down through generations, mostly verbally, which is considered to be tradition [1]. The way of living that encompass this custom persisted since olden times when phytotherapy was one of the main methods for treating people [2]. People had to use what was available in nature throughout the year, regardless of the season. However, the development stage, as well as the conditions during the season, affect the content of bioactive compounds in these medicinal plants. In contemporary studies, the medical potential for most of these plants was confirmed; however, these findings were empirically confirmed by traditional herbalists and healers in the distant past [3].

The Balkan Peninsula, Serbia as well, is inhabited by different nations, and all of them have specific traditions. Nutrition and healing are closely connected to the available resources found in the surrounding nature. Serbian floristic diversity and its ethnobotanical richness is already described [4,5,6,7]. Rtanj Mountain attracts the most attention, especially because of its pyramidal shape, and local people, as well as visitors and tourists, believe in its mystic powers. Rtanj is an isolated mountain in eastern Serbia with a specific ecosystem that is formed on the dominant karst limestone geologic features [7,8]. Its great importance is also indicated by the fact that Rtanj is under governmental protection as a special nature reserve (spread across 4997.17 ha) (“Official Gazette of RS” No. 18/2019).

Traditional harvest of wild medicinal plants is mostly connected to important dates, whether they be dates in regard to the Serbian Orthodox Church or dates connected to important events in nature. The most important day for plant harvesting in Serbia is the Nativity of Saint John the Baptist (7 July according to the Julian calendar and 24 June according to the Gregorian calendar) and it overlaps with the summer solstice. Several customs are tied to that day; however, the most important is that people believe that plants should be collected on that day, which is defined as Biljober (*biljo-ber*, *srb.biljka*, *noun-plant*; *srb.brati*, *verb-picking*). Local people believed that plants collected on this day possess magical properties and stronger healing power.

*Sideritis montana* L., *Teucrium montanum* L., *T. chamaedrys* L., and *Marrubium peregrinum* L. are plants that spontaneously grow in dry meadows and rocky places, such as Rtanj Mt. in eastern Serbia. These plants belong to the Lamiaceae family (*Lamioideae* subfamily) and are used in traditional medicine in this region. Their aboveground parts (*herba*) are collected during the flowering stage. In Serbian agro-ecological conditions, it is usually from June to August. *S. montana* or mountain ironwort (in Serbian “*planinskičistac*”) is usually applied externally for cleaning and healing wounds caused by iron weapons [9]. *T. montanum* or mountain germander (in Serbian “*trava Iva*”) can be used as tea for digestive complaints such as gallbladder problems, for blood purification, and for healing hemorrhoids [10,11]. *T. chamaedrys* or wall germander (in Serbian “*podubica*”) is widely used for curing weaknesses and anemia and for wound cleaning [11]. *M. peregrinum* or horehound (in Serbian “*očajnica*”) is used for regulating the menstrual cycle [7]. However, most of the medicinal uses of these species are limited to folk medicine.

The goal of this research was to examine the composition of volatile components of four species belonging to the family Lamiaceae, subfamily Lamioideae (*S. montana*, *T. montanum*, *T. chamaedrys*, and *M. peregrinum*). Selected plants are characterized by low or trace essential oil content. However, they are widely used in traditional medicines in the regions where they grow. Therefore, the aim of this investigation was to review the ethnomedicinal knowledge and application of selected plants in Serbian and other traditional medicines.

## 2. Results

A total of 34 volatile compounds were detected in *S. montana*, comprising 96.6% (Figure 1a), and the main volatile compound was germacrene D (22.6%), followed by 6,10,14-trimethyl-2-pentadecanone (7.0%), E,E-geranyl linalool (5.5%), and spathulenol (4.6%), as well as trans-β-farnesene (4.3%), trans-caryophyllene (4.0%), abietatriene (3.5%), caryophyllene oxide (3.4%), *δ*-cadinene (3.4%), and two unidentified compounds (4.4% and 3.3%).

In *T. montanum*, a total of 81 volatile compounds were detected comprising 94.7% (Figure 1b). The most dominant among them was the 7-*epi*-*trans*-sesquisabinene hydrate (15.8%) and one unidentified compound (12.2%), followed by *epi*-*α*-cadinol(6.2%), hexadecanoic acid (4.7%), *trans*-caryophyllene (4.2%), *α*-cadinol (3.8%), and limonene (3.4%).

A total of 65 volatile compounds were detected in *T. chamaedrys* comprising 96.0% (Figure 1c). The main compounds were germacrene D(31.8%) and *trans*-caryophyllene (19.7%), followed by 7-*epi*-*α*-selinene (7.2%), *δ*-cadinene (5.5%), *α*-humulene (4.5%), and caryophyllene oxide (3.2%).

In *M. peregrinum*, a total of 64 compounds were detected comprising 94.7% (Figure 1d). The most dominant were *trans*-caryophyllene (32.4%) and *trans*-thujone (25.1%), followed by bicyclogermacrene (5.0%) and two unidentified compounds (3.9% and 3.4%).

## 3. Discussion

### 3.1. Sideritis montana

*S. montana* is a small annual herb with simple or branched upright stems, 20–30 cm tall, covered with thinning long trichomes. Leaves are narrow, ovate-lanceolate with short petiole opposite. Flowers have a yellow corolla, green calyx, and leaf-like bracts, usually with six arranged in verticillasters. After flowering, the corolla becomes red-brown. The flowering period is from May to August [12]. *S. montana* is native to the Mediterranean region, south-western and Central Asia. Nowadays, it can be found in wide regions such as in the Czech Republic, Germany, Poland, Norway, Sweden, Latvia, Estonia, and Lithuania, where it was introduced from Southern Europe [13]. However, *S. montana* is considered a rare and endangered species in Bulgaria, in need of measures for conservation [14]. This species grows in dry and poor meadows, pastures, and rocky and sandy areas [12]. In addition, it is dominant in vegetation of dry pastures and karst in the year after a fire [15]. In Serbian flora, it is recorded as *S. montana* f. *montana* [12].

In the *S. montana* from Rtanj, a total of 34 volatile compounds were detected, comprising 96.6%, and the main volatile compound was germacrene D (22.6%) (Table 1). In Croatia, headspace analysis showed that the main volatile compound in *S. montana* from two localities is germacrene D [16]. In the *S. montana* ssp. *montana* from Italy, a total of 47 volatile compounds were identified (comprising 98.4%). The most abundant compounds were germacrene D (20.8%), bicyclogermacrene (13.3%), and 8,13-abietadien-18-ol (10.2%) [17]. In Turkey, a significant difference in volatile compounds between subspecies was recorded; *S. montana* ssp. *montana* contains 24.6% germacrene D and 10.8% bicyclogermacrene, while subsp. *remota* contains 13.9% bicyclogermacrene and 10.3% germacrene D [18]. Similarly, in the sample from Bulgaria, the main compounds were germacrene D (41.1%) and bicyclogermacrene (10.9%) [19]. As can be seen, germacrene D is the dominant volatile compound in *S. montana*, and variation in content could be attributed to growing locality, variety and extraction type, and analysis method (Table 1).

Apart from the essential oil, *S. montana* is rich in phenolics (caffeic, ferulic, and rosmarinic acid), flavonoids and their derivatives (diosmetin, luteolin-3-O-glucoside, kaempferol-3-O-glucoside, kaempferol-3-O-rutinoside, pomiferin E, and 6-metoxysakuranetin), abietane diterpenoids (sideritins A and B, 9α,13α-epi-dioxyabiet-8(14)-en-18-ol), lignins (paulownin), sesquiterpenoids (3-oxo-α-ionol), phenyl-ethanoid glycosides (verbascoside), phenols (4-allyl-2,6-dimethoxyphenol glucoside), iridoids and their glycosides (ajugol, ajugoside, melittoside), coumarins, terpenes, and sterols (ergosterol, stigmasterol, β-sitosterol), among others [17,20,21,22,23].

*S. montana* is commonly consumed as an herbal tea and it is important in traditional medicine [23]. It is mainly used orally as tea for relieving cough associated with a cold, for reducing fever, against stomach ailments, as an antihysteric, tonic, and stimulant, and is used externally to treat wounds (Table 2). The scientifically proven activities are antioxidant, antimicrobial, anti-inflammatory, smooth muscle-relaxing, anti-proliferative, and cytotoxic activities [17,24,25,26,27,28,29]. These results support the traditional use of *S. montana* for the healing and prevention of many diseases of modern times.

### 3.2. Teucrium montanum

*T. montanum* is a perennial plant with a strong taproot and a prostrate branched shot, 5–25 cm long. Young branches have short internodes and are covered with short grey hairs. Leaves are linear, with a short petiole, dark green on the face, and covered with white hairs on the reverse side. Flowers are white-yellow, grouped in hemispherical inflorescences at the tops of the branches. The flowering period is from June to August [12]. *T. montanum* inhabits thermophilic limestone and serpentine rocks, dry mountain meadows, and edges of forests in Southern Europe and West Asia [35]. This species possesses a pronounced phenotype plasticity manifested through morpho-anatomical and chemical diversity [36]. There are several varieties recorded in Serbian flora: var. *montanum*, var. *pernassicum*, var. *hirsutum*, and var. *skorpilii*, according to differentiation of glandular trichomes [12,37,38].

A total of 81 volatile compounds were detected in *T. montanum* comprising 94.7% (Table 3). The most dominant among them was the 7-*epi*-*trans*-sesquisabinene hydrate (15.8%). A study aimed at determining the composition of *T. montanum* essential oil depending on the geological substrate showed that different chemotypes developed on calcareous and serpentine soils [39]. This study shows that populations from calcareous soils produced and accumulated predominantly aliphatic hydrocarbons, while populations from serpentine soils were characterized by mono- and sesquiterpenes [39]. This can be seen in Table 3, which provides a sample from this study, as well as a review of other studies on chemical composition of volatile components of this plant. *T. montanum* from Italy contained oxygenated sesquiterpenes as the dominant class, with longifolenaldehyde (14.5%), epiglobulol (13.5%), and ledene oxide (12.1%) [40]. *T. montanum* from Croatia contained unsaturated untriacontene (48.4%), followed by nonacosane (17.45%), as the main compounds [41]. Slovak *T. montanum* predominantly contained a sesquiterpene fraction (76.3%), with germacrene D (12.8%), and two unknown oxygenated sesquiterpenes (10.9% and 8.4%), followed by *trans*-caryophyllene (8.0%) [42]. *T. montanum* from Serbia, Jabuka village, contained mainly sesquiterpene hydrocarbons (39.3%) such as *δ*-cadinene and *β*-caryophyllene, as well as oxygenated sesquiterpenes (33.4%) [43]. A similar composition was also obtained in Serbia, Jadovnik Mt, with δ-cadinene (17.2%) and β-selinene (8.2%) [44]. Main constituents of *T. montanum* from Montenegro were germacrene D (15.0%), α-pinene (12.4%), and β-eudesmol (10.1%) [45].

Apart from the essential oil, *T. montanum* contains polyphenolic compounds such as phenolic acids (hydroxyl derivatives of benzoic and cinnamic acids), phenylethanoid glycoside (verbascoside and echinacoside), flavonoids and their glycosides (cirsiliol, luteolin, apigenin, cirsimaritin, rutin, naringin, epicatechin, catechin, luteolin-7-O-rutinoside, luteolin-7-O-glucoside, quercetin-3-O-rutinoside, and diosmetin-7-O-rutinoside), coumarins, diterpenoids (19-acetylgnaphalin, montanin B,D,E, and teubotrin), and triterpenes [46,47,48,49,50].

*T. montanum* is widely used in traditional medicines in many Balkans countries [51] but predominantly in Bosnia and Herzegovina, Serbia, Montenegro, and Kosovo. It is used for treating a wide range of aliments, such as digestive complaints (abdominal pain, constipation, liver damage and gallstones, spasm relief, for improving appetite, etc.), for immune system strengthening, as a tonic, for blood purification, against respiratory disorders such as tuberculosis, as an antipyretic, and for treating rheumatism and skin problems (Table 4). In some regions (Herzegovina, Kosovo, and Croatia), this plant is consumed as a tea, eaten as a dish, or added to alcoholic beverages (alcoholic beverage with herbs, traditionally called “*travarica*”) [52,53,54,55]. The scientifically proven activities of *T. montanum* are as follows: antitumor, cytotoxic, antioxidant, and antibacterial activities. According to this review, *T. montanum* can be regarded as a promising candidate to be a natural plant source of effective biological compounds, as a supplement in the food industry, as well as for therapeutic use [56].

### 3.3. Teucrium chamaedrys

*T. chamaedrys* is a small *shrub* with a woody-based root system, and it develops underground stolones. The stem is upright and spreading, 10–30 cm high. Leaves have short petiole, are broad with many rounded lobes and a broad rounded tip similar to common oak (in Serbian “*dub*” because of leaf similarity with *Quercus robur*, the common Serbian name for *T. chamaedrys*is “*podubica*”, i.e., like oak). Pink flowers appear during summer (from June to August) [12]. The plant inhabits rocky limestone areas, dry mountain meadows and pastures, and edges of sparse oak and pine forests up to 1000 m above sea level in Central Europe, the Mediterranean region, and Western Asia [35]. In Serbian flora, it is recorded as var. *glanduliferum* and var. *chamaedrys* with two forms: f. *chamaedrys* and f. *viride* [12].

A total of 65 volatile compounds were detected in *T. chamaedrys* comprising 96.0%, and the main compounds were germacrene D (31.8%) and *trans*-caryophyllene (19.7%) (Table 5). The main constituents of *T. chamaedrys* essential oil from Turkey were germacrene D (32.1%), *trans*-caryophyllene (14.2%), *δ*-cadinene (13.1%), and bicyclogermacrene (6.7%) [67]. *T. chamaedrys* ssp. *syspirense* from Turkey contains *trans*-caryophyllene (18.2%), germacrene D (10.8%), carvacrol (9.5%), and *α*-humulene (6.4%) as dominant constituents in its essential oil [68]. The main compounds in *T. chamaedrys* from Corsica were *trans*-caryophyllene (29.0%) and germacrene D (19.4%), followed by *α*-humulene (6.8%) and *δ*-cadinene (5.4%). The sample from Sardinia contained *trans*-caryophyllene (27.4%) and germacrene D (13.5%); however, it also contained caryophyllene oxide (12.3%) and *α*-humulene (6.5%) as dominant compounds [69]. The main constituents in *T. chamaedrys* from Montenegro were *trans*-caryophyllene (26.9%) and germacrene D (22.8%) [45].

Apart from essential oil, *T. chamaedrys* contains iridoids (harpagide), neoclerodane diterpenoids, flavonoids and their derivatives (apigenin, cirsiliol, cirsimaritin, luteolin-7-O-rutinoside, luteolin7-O-glucoside, quercetin-3-O-rutinoside, apigenin-7-O-rutinoside, apigenin-7-O-glucoside, and diosmetin-7-O-rutinoside), phenyl-ethanoid glycosides (forsythoside B, verbascoside, samioside, and alyssonoside), phenolic compounds (hydroxycinamic acid derivatives), triterpenoids, and steroids [48,71,72].

*T. chamaedrys* is one of the most popular traditional remedies in the Balkans, used as tea in everyday nutrition [53], as well as for treating many disorders (Table 6). It is used in Turkey, Serbia, Kosovo, and Bosnia and Herzegovina for gastrointestinal aliments, such as spasm relief, liver, spleen and gall aliments, diarrhea, loss of appetite, stomachache, hemorrhoids, and against ulcers, respiratory ailments including bronchitis, tuberculosis, fever, as well as vaginal infections, kidney pain, chronic inflammation of the mucous membranes in the eyes and nose, toothache, and many others. There are only a few scientifically proven activities of *T. chamaedrys*: antioxidant, antimicrobial, thyrosinase inhibitory effect, and cytotoxic activities. However, hepatotoxic effects have been reported for *T. chamaedrys* because this plant contains neoclerodanediterpenes [73]. Therefore, controlled application of this plant is necessary.

### 3.4. Marrubium peregrinum

*M. peregrinum* is a perennial herbaceous plant. Above-ground parts are gray, densely covered with short hairs and trichomes. Stem is erect, 30–60 cm high, branched in the upper half. The lower leaves are ovate, narrowed at the base into a petiole, while other leaves are elongated to lanceolate and saw-toothed. It has several flowers, usually 6–12 grouped in loose spherical inflorescences. It blooms during July and August. As a typical Pontic-Mediterranean species, it is widespread in Central Europe, the Balkan Peninsula, and Asia Minor. Its habitats are dry pastures, rocky meadows, loams, and sandy soils [12]. It usually grows at low altitudes, generally below 1000 m [81].

A total of 64 compounds were detected in *M. peregrinum* comprising 94.7%, and *trans*-caryophyllene (32.4%) and *trans*-thujone (25.1%) were dominant (Table 7). Investigation of *M. peregrinum* essential oils from three different locations in Vojvodina province (north part of Serbia) shows that they contain sesquiterpene hydrocarbons: trans-caryophyllene (13.2–18.0%), bicyclogermacrene (6.4–9.8%), and germacrene D (6.8–9.1%) [82]. The essential oil of *M. peregrinum* growing wild in Greece contains *cis*- and *trans*-*β*-farnesene as dominant compounds (12.0–16.5% and 21.5–24.2%, respectively, depending on the population) [81]. In *M. peregrinum* essential oil from Slovakia, a total of 16 compounds were identified comprising 98.1%. Dominant compounds were *trans*-caryophyllene (31.3%), germacrene D (28.1%), and bicyclogermacrene (15.3%) [83].

Compounds isolated from various extracts of *M. peregrinum* are β-sitosterol, labdane diterpenoids (peregrinin, preperegrinin, peregrinol, marrubiin, premarrubiin, cyllenin, and 15-epi-cyllenin A), and phenolic compounds flavone aglycones, flavone glycosides, coumaroylated flavone glycosides, and acteoside-related phenyl-ethanoids (ladanein, 6-hydroxy-5,7,4′-trimetoxyflavone and 5,6,7,4′-tetramethoxyflavone, apigenin, kaemferol, apigenin-7-glucoside, luteolin-7-glucoside, and acteoside) [84].

Aerial parts of *M. peregrinum* possess bitter principles similar to *M. vulgare*. However, this plant is rarely used in traditional medicine (Table 8). It is recorded only in Serbian ethnopharmacology, in the region of eastern Serbia (Mt. Rtanj and Svrljiški Timok gorge), as well as in Vojvodina province (northeastern part of Serbia; Deliblato Sands) [7,74,85]. In Bulgaria, it is recorded as a medicinal and spice plant but without a detailed explanation, as well as a plant for making garden brooms [86,87]. There are few studies dealing with the biological properties of *M. peregrinum*, which show antioxidant [82,88] and antimicrobial activities [89]. This plant can be promising as a natural source of antioxidants and antimicrobial agents.

## 4. Materials and Methods

### 4.1. Plant Material

Selected non-aromatic medicinal plants (*Lamiaceae, Lamioideae*) were collected at Rtanj Mountain (Southern Carpathians chain, East Serbia), on 7 July 2019 (St. John the Baptist Day). All investigated plants were in the flowering stage (Figure 2), harvested manually by gardening scissors, gathered in bouquets and dried hung upside down in a shaded, well-ventilated place for one week. The selected plants, *Sideritis montana* L. (2–403), *Teucrium montanum* L. (2–1405), *T. chamaedrys* L. (2–1404), and *Marrubium peregrinum* L. (2–1409), were determined and deposited at Buns Herbarium, University of Novi Sad, Serbia.

### 4.2. Essential Oil Extraction and Analysis

The essential oils were isolated by hydrodistillation from dry above-ground plant parts using a Clevenger-type apparatus using 30.0 g finely ground plant material and 500 mL of water during 2 h. There was a small quantity of extracted essential oils (less than 0.1 mL) in all four samples, so it was additionally applied liquid–liquid extraction with hexane as the solvent.

The GC-FID and GC-MS analysis was performed using an Agilent 7890 gas chromatograph coupled with an Agilent 5975C MSD and flame ionization detector on a nonpolar HP-5MS fused-silica capillary column Agilent 19091S-433 (conditions were mimicked from Adams [90] and described in detail by Acimovic et al. [91]). The identification of constituents was carried out based on the retention index and by comparison with reference spectra (Wiley 7, NIST 17, and retention-time-locked Adams 4 databases) using the Automated Mass Spectral Deconvolution and Identification System (Amdis 32 ver 2.73) and NIST search ver. 2.3. The relative percentage of the oil constituents was expressed as percentages by FID peak-area normalization.

### 4.3. Data Collection

A bibliographic search on Science Direct, PubMed, and Google Scholar was conducted to collect information about the chemical profiles of volatile components of selected non-aromatic medicinal plants, their traditional applications in folk medicine in Serbia and neighborhood regions (Kosovo, Montenegro, Bosnia and Herzegovina, and Bulgaria) and wider (Spain, Algeria, and Turkey), as well as modern phytotherapy approaches according to bioactivity tests (antioxidant, antimicrobial, anti-inflammatory, antiproliferative and cytotoxic activity, etc.).

## 5. Conclusions

Ironwort (*Sideritis montana* L.), mountain germander *(Teucrium montanum* L.), wall germander (*Teucrium chamaedrys* L.), and horehound (*Marrubium peregrinum* L.) belong to the Lamiaceae family and subfamily Lamioideae. Essential oil content in selected medicinal plants is low (characteristic of the subfamily), but essential oil composition could be useful as an achemotaxonomic marker for the selected location, taking into account the wide range of distribution. In the selected plants from Serbia (Rtanj Mt), the sesquiterpenes were the most abundant class of volatile components: germacrene D (22.6%) in ironwort, *trans*-sesquisabinene hydrate (15.8%) in mountain germander, germacrene D (31.8%) and *trans*-caryophyllene (19.7%) in wall germander, and *trans*-caryophyllene (32.4%) and *trans*-thujone (25.1%) in horehound. However, these plants contain significant numbers of other bioactive compounds such as phenols, flavonoids, diterpenes and diterpenoids, iridoids and their glycosides, coumarins, terpenes, and sterols, which contribute to their biological potential.

In traditional medicine, ironwort is used to relieve coughs associated with cold coughs, as an antipyretic, for stomach ailments, as a digestive infusion, tonic, and stimulant, as well as an antihysteria and wound-healing agent. Modern phytotherapy reveals that it has antioxidant, antimicrobial, anti-inflammatory, antiproliferative, and cytotoxic effects, as well as the effect of relaxing smooth muscles. In traditional medicine, mountain germander is mainly used for the treatment of digestive problems, strengthening of the immune system, blood purification, antipyretic, respiratory diseases, rheumatism, and skin problems, while modern science shows that it has antioxidant, antibacterial, antitumor, and cytotoxic effects. In traditional medicine, the wall germander is used for digestive problems, respiratory system diseases, heart diseases, and other types of pain and inflammation. However, modern science confirms its antioxidant, antimicrobial, tyrosinase inhibitory, and cytotoxic effects. Traditional Serbian medicine recommends horehound against menstrual problems, anemia, hemorrhoids, and digestive problems, and as a tonic, excitant, reliever, and secretion stimulant, and for the treatment of respiratory tract problems, arrhythmias, and general weakness. However, there are only scientifically confirmed antioxidant and antimicrobial activities for this plant.

Considering the growing popularity of traditional systems of healing, ironwort, mountain germander, wall germander, and horehound could be used as natural agents for promoting health, as well as sources of bioactive compounds for the pharmaceutical and food industries based on traditional knowledge and approved by a modern phytotherapy approach. In addition, sources of raw material could be significant to biological standardization of herbal preparations. Future research should be focused on herbal or polyherbal formulations, investigation of their biological activity, as well as their application in everyday life as commercial products.

## Figures and Tables

**Figure 1 molecules-28-04611-f001:**
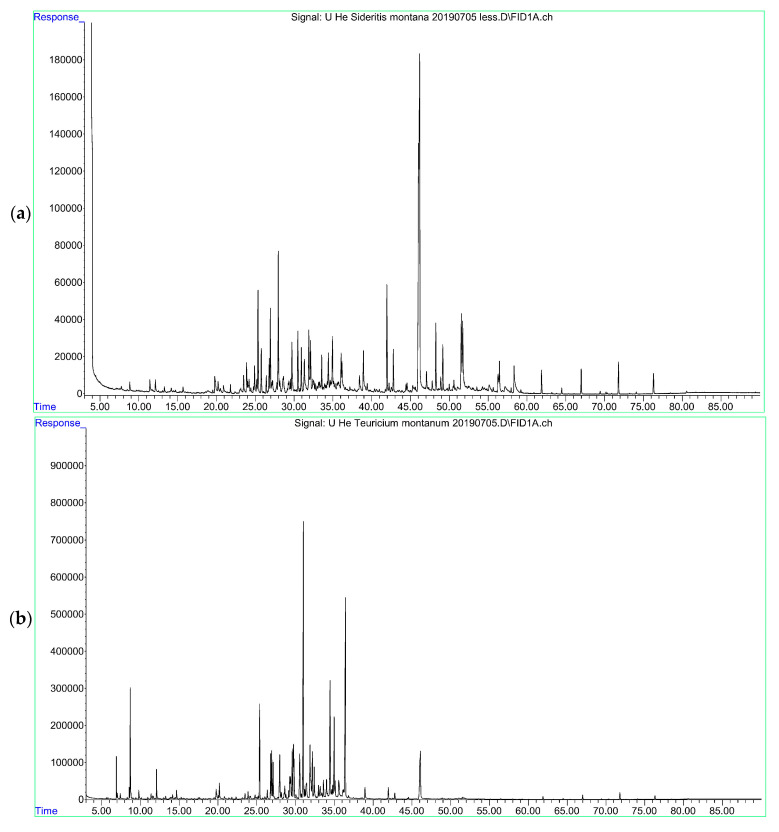
GC-MS chromatograms: (**a**) *Sideritis montana*; (**b**) *Teucrium montanum*; (**c**) *Teucrium chamaedrys*; (**d**) *Marrubium peregrinum*.

**Figure 2 molecules-28-04611-f002:**
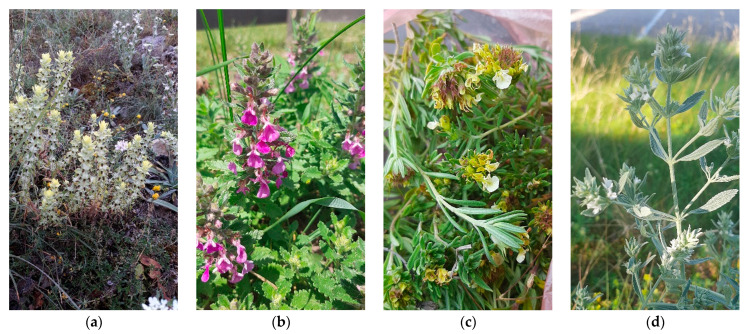
Selected non-aromatic medicinal plants (*Lamiaceae, Lamioideae*) collected at Rtanj Mt: (**a**) *Sideritis montana*; (**b**) *Teucrium montanum*; (**c**) *Teucrium chamaedrys*; (**d**) *Marrubium peregrinum*.

**Table 1 molecules-28-04611-t001:** Volatile compounds in *Sideritis montana* from Rtanj Mt, Serbia (this study—TS) and references data.

No	Chemical Compound	RI_exp_	RI_lit_	Rtanj, Serbia (TS)	Croatia, Ježević [16]	Croatia, Mosor [16]	Italy, Capolapiaggia Mt [17]	Turkey, Kirklareli [18]	Turkey, Eskisehir [18]	Bulgaria [19]
1	1,8-Cineole	1028	1026	0.4	-	-	-	0.3	0.2	-
2	*trans*-Thujone	1114	1112	0.4	-	-	-	-	-	-
3	Camphor	1139	1141	0.3	-	-	-	-	-	-
4	Borneol	1160	1165	0.2	-	-	-	-	-	-
5	*α*-Copaene	1370	1374	1.1	1.7	2.0	1.3	1.0	2.8	1.2
6	*β*-Bourbonene	1379	1387	1.8	-	-	1.2	1.1	3.0	-
7	*β*-Elemene	1386	1389	0.5	-	-	0.8	0.4	0.5	-
8	Dodecanal	1402	1408	0.7	-	-	-	-	-	-
9	*trans*-Caryophyllene	1412	1417	4.0	11.9	6.6	3.2	4.0	-	8.8
10	*β*-Copaene	1423	1430	0.5	-	-	0.3	-	-	-
11	*α*-Humulene	1448	1452	0.9	-	-	0.3	0.4	0.1	1.6
12	*trans*-*β*-Farnesene	1451	1454	4.3	4.8	3.6	3.2	7.2	2.9	-
13	Germacrene D	1475	1484	22.6	23.2	17.0	20.8	24.6	10.3	41.1
14	NI-1	1492	/	2.4	-	-	-	-	-	-
15	*γ*-Cadinene	1509	1513	0.9	3.3	3.4	-	0.1	0.4	-
16	*δ*-Cadinene	1518	1522	3.4	7.6	8.9	1.8	1.5	3.0	1.2
17	NI-2	1536	/	4.4	-	-	-	-	-	-
18	Spathulenol	1571	1577	4.6	0.2	0.5	0.8	1.1	4.8	-
19	Caryophyllene oxide	1576	1582	3.4	-	-	0.1	-	0.7	-
20	NI-3	1611	/	2.5	-	-	-	-	-	-
21	*epi*-*α*-Cadinol (=*τ*-Cadinol)	1634	1638	2.7	-	-	-	0.5	1.0	1.8
22	NI-4	1648	/	3.3	-	-	-	-	-	-
23	*α*-Bisabolone oxide A	1677	1684	1.4	-	-	-	-	-	-
24	Germacra-4(15),5,10(14)-trien-1-*α*-ol	1680	1685	1.3	-	-	0.1	-	-	-
25	6,10,14-trimethyl-2-Pentadecanone	1841	1847	7.0	-	-	-	-	-	-
26	Manool oxide	1988	1987	1.5	-	-	-	-	-	-
27	E,E-Geranyl linalool	2026	2026	5.5	-	-	-	-	-	-
28	Abietatriene	2055	2055	3.5	-	-	0.1	-	-	-
29	Tricosane	2300	2300	2.3	-	-	-	0.3	-	-
30	Pentacosane	2500	2500	1.8	-	-	0.5	-	-	-
31	Hexacosane	2600	2600	0.9	-	-	-	-	-	-
32	Heptacosane	2700	2700	2.2	-	-	0.9	-	-	-
33	Nonacosane	2900	2900	2.5	-	-	1.4	-	-	-
34	Untriacontane	3100	3100	1.4	-	-	0.2	-	-	-
	Other *			-	36.6	44.5	61.4	49.3	56.2	32.2
	Oxygenated monoterpenes			1.3						
	Sesquiterpene hydrocarbons			40.0						
	Oxygenated sesquiterpenes			13.4						
	Diterpenes hydrocarbons			3.5						
	Oxygenated Diterpenes hydrocarbons			7.0						
	Other			18.8						
	Total Identified			96.6	89.3	86.4	98.4	91.9	86.0	87.9

RI_exp_—retention indices experimentally obtained by C_8_–C_32_ *n-*alkanes series; RI_lit_—retention indices literally (RI library Adams4 and Nist webbook); * sum of compounds not detected in sample from this study. NI-1: 161(100), 121(66), 120(38), 81(35), 67(31), 105(29), 204(29), 162(25), 106(23), 91(21). NI-2: 107(100), 132(50), 91(34), 105(34), 125(30), 119(29), 133(29), 122(28), 41(23), 93(22). NI-3: 109(100), 124(97), 81(94), 95(74), 41(65), 82(58), 107(55), 67(54), 93(48), 55(46). NI-4: 43(100), 161(89), 95(85), 105(83), 81(79), 121(74), 93(62), 204(61), 41(60), 91(58).

**Table 2 molecules-28-04611-t002:** Traditional use of *Sideritis montana* and scientifically proven activities.

Traditional Use			Scientifically Proven Activities		
Country/Ailments Treated	Part Used/Preparation/Administration	Reference	Activity	Form	Reference
Turkey: cough, stomach aliments	herb (aerial parts)/infusion/internally	[30,31]	Antioxidant	ethanol, methanol, butyl methyl ether, acetone, ethyl acetate, butanol and hexane extracts, essential oil	[17,24,25,28,29]
Algeria: febrifuge, tonic, stimulant, anti-hysterical	whole plant/ns/internally	[32]	Antimicrobial	methanol, acetone and ethyl acetate extracts, essential oil	[25,26]
Serbia: wound healing	herb (aerial parts)/decoction, infusion/internally	[9]	Smooth muscle-relaxing	methanol extract	[27]
Bulgaria: for relief of cough associated with cold	ns	[33]	Anti-proliferative (cervical cancer) and cytotoxic (melanoma, breast adenocarcinoma and human colon cancer)	ethanol, methanol, hexane and ethyl acetate extracts, essential oil	[17,22]
Spain: digestive	ns/infusion/internally	[34]	Anti-inflammatory	methanol extract	[28]

ns—not specified.

**Table 3 molecules-28-04611-t003:** Volatile compounds in *Teucrium montanum* from Rtanj Mt, Serbia (this study—TS) and references data.

No	Chemical Compound	RI_exp_	RI_lit_	Rtanj, Serbia (TS)	Sicily, Italia [40]	Trilj, Croatia [41]	Slovak Karst, Slovakia [42]	Jabuka, Srbija [43]	Jadovnik, Serbia [44]	Orjen, Montenegro [45]
1	Sabinene	969	969	1.1	0.5	0.4	0.6	-	tr	0.8
2	*β*-Pinene	973	974	0.2	-	0.1	2.2	1.6	-	4.8
3	Myrcene	987	988	0.2	-	-	1.3	0.2	-	0.3
4	*α*-Terpinene	1014	1014	0.1	-	-	tr	-	tr	-
5	*p*-Cymene	1021	1020	0.4	-	-	tr	0.2	0.7	0.2
6	Limonene	1025	1024	3.4	-	-	0.4	1.0	-	1.8
7	1,8-Cineole	1028	1026	0.2	-	-	-	-	-	-
8	*trans*-*β*-Ocimene	1044	1044	0.1	-	-	-	-	-	0.5
9	*γ*-Terpinene	1054	1054	0.3	-	-	tr	-	0.4	-
10	Terpinolene	1085	1086	0.1	-	-	0.1	-	-	-
11	Linalool	1097	1095	0.3	-	-	0.2	0.5	-	-
12	*n*-Nonanal	1102	1100	0.1	-	-	-	-	-	-
13	*cis*-Thujone	1103	1101	0.1	-	-	-	-	-	-
14	*trans*-Thujone	1114	1112	1.1	-	-	-	-	-	-
15	*trans*-Pinocarveol	1134	1135	0.1	-	-	0.2	-	-	-
16	Camphor	1139	1141	0.1	-	-	-	-	-	-
17	Sabina ketone	1152	1154	0.1	-	-	-	-	-	-
18	*trans*-Pinocamphone	1155	1158	0.1	-	-	-	-	-	-
19	Borneol	1160	1165	0.2	-	-	tr	-	-	-
20	Terpinen-4-ol	1170	1174	0.4	-	-	0.1	0.1	-	-
21	*α*-Terpineol	1184	1186	0.1	-	-	0.3	0.2	-	-
22	Cumin aldehyde	1234	1238	0.1	-	-	-	-	-	-
23	Carvone	1237	1239	0.1	-	-	tr	0.2	-	-
24	Bornyl acetate	1280	1287	0.1	-	-	-	0.6	-	-
25	Thymol	1285	1289	0.4	-	-	-	-	-	-
26	Theaspirane	1292	1301	0.1	-	-	-	0.2	-	-
27	Carvacrol	1295	1298	0.7	-	-	-	-	-	-
28	*p*-Mentha-1,4-dien-7-ol	1323	1325	0.1	-	-	-	-	-	-
29	*δ*-Elemene	1331	1335	0.1	-	-	-	-	-	-
30	*α*-Copaene	1370	1374	0.2	2.3	-	0.4	0.6	-	0.5
31	*β*-Bourbonene	1379	1387	0.3	-	0.5	1.1	tr	-	1.9
32	*β*-Cubebene	1384	1387	0.1	-	-	-	tr	-	0.3
33	*β*-Elemene	1386	1389	0.1	-	-	0.4	0.6	-	-
34	Sesquithujene	1399	1405	0.2	-	-	-	0.1	-	-
35	*cis*-*α*-Bergamotene	1409	1411	0.1	-	-	tr	-	-	-
36	*trans*-Caryophyllene	1412	1417	4.2	1.9	tr	8.0	5.1	4.4	6.9
37	*trans*-*α*-Bergamotene	1429	1432	0.1	2.1	-	1.4	0.7	1.1	-
38	*cis*-*β*-Farnesene	1437	1440	0.5	-	0.9	0.1	1.8	-	-
39	*α*-Humulene	1448	1452	2.1	2.5	-	1.4	3.1	-	1.7
40	*trans*-*β*-Farnesene	1451	1454	2.2	-	1.0	2.0	1.5	-	0.2
41	9-*epi*-*trans*-Caryophyllene	1455	1464	1.8	-	-	-	-	-	-
42	*α*-Acoradiene	1458	1464	0.1	-	-	-	-	-	-
43	*γ*-Muurolene	1471	1478	0.1	-	-	-	1.1	-	-
44	*γ*-Curcumene	1474	1481	0.4	-	-	-	0.7	3.2	-
45	Germacrene D	1475	1484	2.2	-	3.7	12.8	0.2	-	15.0
46	*β*-Selinene	1481	1489	0.3	-	-	0.4	-	8.2	1.2
47	*trans*-Muurola-4(14),5-diene	1489	1493	0.2	-	-	0.3	0.5	-	-
48	Bicyclogermacrene	1491	1500	0.6	-	0.4	3.1	-	-	3.5
49	*α*-Muurolene	1495	1500	0.2	-	-	0.3	2.3	1.7	0.3
50	*β*-Bisabolene	1503	1505	0.4	-	tr	-	0.5	0.7	-
51	*β*-Curcumene	1507	1514	0.9	-	-	-	0.6	-	-
52	*γ*-Cadinene	1509	1513	1.5	-	-	-	3.6	-	4.1
53	NI-1	1514		2.1	-	-	-	-	-	-
54	*δ*-Cadinene	1518	1522	2.2	1.8	-	-	8.1	17.2	4.5
55	NI-2	1520	/	1.7	-	-	-	-	-	-
56	*cis*-Sesquisabinene hydrate (IPP vs. OH)	1537	1542	3.0	-	1.8	0.2	1.9	-	-
57	*7-epi-trans*-Sesquisabinene hydrate	1549	1543	15.8	-	0.5	1.1	-	-	-
58	NI-3	1570	/	2.5	-	-	-	-	-	-
59	Caryophyllene oxide	1576	1582	2.0	2.8	-	2.5	2.0	-	2.6
60	NI-4	1584	/	1.4	-	-	-	-	-	-
61	Humulene epoxide II	1603	1608	0.4	-	-	0.3	0.2	-	-
62	*epi*-Cedrol	1607	1618	0.3	-	-	-	-	-	-
63	10-*epi*-*γ*-Eudesmol	1611	1622	0.9	-	-	-	-	-	-
64	*α*-Acorenol	1622	1630	0.9	-	-	-	-	-	-
65	*epi*-*α*-Cadinol (=*τ*-Cadinol)	1634	1638	6.2	-	0.5	-	-	3.1	-
66	*α*-Muurolol (=Torreyol)	1640	1644	0.2	-	-	0.4	0.7	3.9	-
67	*β*-Eudesmol	1644	1649	0.4	-	-	-	-	-	10.1
68	*α*-Cadinol	1649	1652	3.8	-	-	1.8	3.5	-	3.5
69	7-*epi*-*α*-Eudesmol	1652	1662	0.7	-	-	-	-	-	-
70	*epi*-*β*-Bisabolol	1665	1670	0.8	-	-	0.1	0.9	-	-
71	*β*-Bisabolol	1666	1674	0.6	3.9	-	-	-	-	-
72	*α*-Bisabolol	1681	1685	0.4	-	-	-	-	-	-
73	NI-5	1687	/	12.2	-	-	-	-	-	-
74	Tetradecanoic acid	1756	1761	0.7	-	-	-	-	-	-
75	6,10,14-trimethyl-2-Pentadecanone	1841	1847	0.6	-	-	-	-	-	-
76	5E,9E-Farnesyl acetone	1917	1913	0.1	-	-	-	-	-	-
77	Hexadecanoic acid	1959	1959	4.7	-	-	-	-	-	-
78	Pentacosane	2500	2500	0.1	-	1.2	tr	-	-	-
79	Heptacosane	2700	2700	0.2	-	3.9	tr	-	-	-
80	Nonacosane	2900	2900	0.3	-	17.5	tr	-	-	-
81	Untriacontane	3100	3100	0.2	-	0.2	tr	-	-	-
	Other *			-	78.1	61.6	56.6	53.3	53.4	33.4
	Monoterpene hydrocarbons			5.9						
	Oxygenated monoterpenes			4.3						
	Sesquiterpene hydrocarbons			21.1						
	Oxygenated sesquiterpenes			36.5						
	Other			7.0						
	Total Identified			94.7	95.9	94.2	98.8	98.4	98.0	98.1

RI_exp_—retention indices experimentally obtained by C_8_–C_32_ *n-*alkanes series; RI_lit_—retention indices literally (RI library Adams4 and Nist webbook); * sum of compounds not detected in sample from this study. NI-1: 81(100), 121(75), 93(72), 109(61), 41(54), 69(52), 55(51), 67(45), 95(42), 43(41). NI-2: 109(100), 81(75), 93(62), 121(54), 67(49), 95(46), 55(45), 83(44), 123(43), 136(40). NI-3: 161(100), 81(69), 105(58), 119(42), 91(35), 121(33), 204(32), 93(30), 43(25), 79(25). NI-4: 119(100), 93(75), 69(60), 91(48), 41(47), 105(40), 121(38), 77(31), 79(29), 161(29). NI-5: 161(100), 84(85), 81(76), 105(64), 41(50), 91(44), 119(44), 55(43), 93(43), 109(41).

**Table 4 molecules-28-04611-t004:** Traditional use of *Teucrium montanum* and scientifically proven activities.

Traditional Use			Scientifically Proven Activities		
Country/Ailments Treated	Part Used/Preparation/Administration	Reference	Activity	Form	Reference
Bosnia and Herzegovina: digestive complains, liver and gall aliments (gallstones), spasm relief, blood purification, pulmonary aliments, rheumatism	aerial parts/infusion/internally	[10,57,58,59,60]	Antitumor (chronic myelogenous leukemia, cervix adenocarcinoma)	methanol extract	[56]
Serbia: digestive complains, abdominal pain, constipation, immune system strengthening, tonic, improving appetite, respiratory disorders, antipyretic, tuberculosis	aerial parts/infusion/internally; bath soak, inhalation/externally	[4,5,6,7,11,61,62]	Cytotoxic (cervix carcinoma, rhabdomyosarcoma and murine fibroblast cells)	ethanol extract	[63]
Montenegro: respiratory and gastrointestinal disorders	aerial parts/infusion/internally	[64]	Antibacterial	methanol, petroleum ether, chloroform, ethyl acetate and n-butanol extracts, essential oil	[44,65]
Kosovo: skin problems	leaves/infusion/externally	[66]	Antioxidant	petroleum ether, chloroform, ethyl acetate, n-butanol and subcritical water extracts	[49,65]

**Table 5 molecules-28-04611-t005:** Volatile compounds in *Teurium chamaedrys* from Rtanj Mt, Serbia (this study—TS) and references data.

No	Chemical Compound	RI_exp_	RI_lit_	Rtanj, Serbia (TS)	Moldova [70]	Turkey [67]	Iran [68]	Corsica, France [69]	Sardinia, Italy [69]	Orjen, Montenegro [45]
1	*α*-Pinene	931	932	0.2	1.7	0.2	1.0	1.0	4.4	5.3
2	1-Octen-3-ol	975	974	0.8	-	-	1.7	1.4	0.2	3.7
3	Limonene	1025	1024	0.1	-	-	0.4	-		1.4
4	1,8-Cineole	1028	1026	0.2	-	-	0.3	-		0.2
5	Linalool	1097	1095	0.5	-	-	3.7	0.8	0.1	-
6	1-Octen-3-yl acetate	1111	1110	0.1	-	-	-	0.1	-	-
7	*trans*-Thujone	1114	1112	0.8	-	-	0.5	-		-
8	*trans*-Pinocarveol	1134	1135	0.1	-	-	tr	0.1	tr	-
9	Camphor	1139	1141	0.2	-	-	-	-		-
10	*trans*-Pinocamphone	1155	1158	0.1	-	-	-	-		-
11	Borneol	1160	1165	0.1	-	-	tr	-		-
12	Terpinen-4-ol	1170	1174	0.1	-	-	tr	tr	tr	-
13	*α*-Terpineol	1184	1186	0.1	-	-	0.2	0.2	tr	-
14	Myrtenal	1195	1195	0.1	-	-	0.2	0.1	-	-
15	Linalool acetate	1255	1254	0.1	-	-	-	-		-
16	Isobornyl acetate	1287	1283	0.1	-	-	-	-		-
17	Thymol	1291	1289	0.2	-	-	1.0	-		-
18	Dihydroedulan II	1293	1290	0.1	-	-	-	tr	tr	-
19	Carvacrol	1300	1298	0.2	-	-	9.5	-		-
20	*δ*-Elemene	1336	1335	0.3	-	-	-	-		-
21	*α*-Cubebene	1349	1345	0.1	1.8	-	-	tr	-	0.3
22	*α*-Copaene	1375	1374	0.5	0.8	-	0.3	0.3	0.2	0.5
23	*β*-Bourbonene	1383	1387	1.9	1.7	2.4	3.2	3.1	3.0	2.2
24	*β*-Cubebene	1389	1387	0.2	-	0.2	-	-		0.7
25	*β*-Elemene	1391	1389	0.3	-	-	-	-		-
26	*α*-Gurjunene	1408	1409	0.1	-	0.3	-	-		-
27	*trans*-Caryophyllene	1418	1417	19.7	41.0	14.2	18.2	29.0	27.4	26.9
28	*β*-Copaene	1428	1430	0.8	-	0.6	-	0.6	0.5	-
29	*trans*-*α*-Bergamotene	1434	1432	0.1	-	-	3.3	0.1	tr	0.5
30	6,9-Guaiadiene	1442	1442	0.3	-	-	-	-		-
31	*α*-Humulene	1452	1452	4.5	-	1.8	6.4	6.8	6.5	6.7
32	*trans*-*β*-Farnesene	1456	1454	0.3	-	4.3	2.5	4.4	1.9	0.6
33	*allo*-Aromadendrene	1460	1458	0.9	0.8	-	0.2	0.6	0.7	-
34	*cis*-Muurola-4(14),5-diene	1463	1465	0.1	-	-	-	-		-
35	*γ*-Muurolene	1479	1478	0.4	-	-	-	-		-
36	Germacrene D	1482	1484	31.8	22.1	32.1	10.8	19.4	13.5	22.8
37	*β*-Selinene	1486	1489	0.4	-		-	-		-
38	Valencene	1492	1498	1.6	0.8	-	-	-		-
39	Bicyclogermacrene	1496	1500	2.3	1.7	6.7	2.0	1.6	0.9	2.2
40	*α*-Muurolene	1500	1500	0.3	-	-	0.8	0.3	0.1	-
41	*β*-Bisabolene	1503	1505	0.8	-	-	1.0	1.6	0.4	1.3
42	*γ*-Cadinene	1514	1513	0.3	-	0.2	1.0	0.1	0.3	1.1
43	7-*epi*-*α*-Selinene	1518	1520	7.2	-	-	-	0.1	0.1	-
44	*δ*-Cadinene	1523	1522	5.5	-	13.1	3.1	5.4	1.7	3.1
45	*trans*-Cadina-1,4-diene	1532	1533	0.1	-	-	-	0.2	0.1	-
46	Elemol	1548	1548	0.1	-	-	-	-		0.4
47	1-*nor*-Bourbonanone	1560	1561	0.1	-	-	-	-		-
48	*β*-Calacorene	1565	1564	0.1	-	-	-	-		-
49	Spathulenol	1576	1577	1.0	-	-	2.8	-		-
50	Caryophyllene oxide	1581	1582	3.2	2.2	1.2	4.8	3.2	12.3	5.5
51	Humulene epoxide II	1607	1608	0.6	-	-	-	0.6	2.4	-
52	Muurola-4,10(14)-dien-1-*β*-ol	1626	1630	0.2	-	-	-	-		-
53	*epi*-*α*-Muurolol (=*τ*-Muurolol)	1640	1640	0.7	-	-	-	0.3	0.2	-
54	*α*-Cadinol	1649	1652	1.2	-	-	1.4	0.8	0.1	0.7
55	14-hydroxy-9-*epi*-*trans*-Caryophyllene	1669	1668	0.5	-	-	-	-		-
56	*epi*-*β*-Bisabolol	1682	1670	0.1	-	-	2.1	-		-
57	Germacra-4(15),5,10(14)-trien-1-*α*-ol	1685	1685	0.9	-	-	-	-		-
58	6,10,14-trimethyl-2-Pentadecanone	1843	1847	0.5	-	-	-	0.2	2.1	-
59	5E,9E-Farnesyl acetone	1918	1913	0.2	-	-	-	-		-
60	Phytol	2116	2122	0.7	-	-	-	1.8	0.4	-
61	Tricosane	2300	2300	0.1	-	0.8	-	tr	tr	-
62	Pentacosane	2500	2500	0.3	-	-	-	-		-
63	Heptacosane	2700	2700	0.2	-	-	-	-		-
64	Nonacosane	2900	2900	0.2	-	-	-	-		-
65	Untriacontane	3100	3100	0.1	-	-	-	-		-
	Other *			-	24.3	12.7	16.4	8.5	12.5	11.4
	Monoterpene hydrocarbons			0.3						
	Oxygenated monoterpenes			2.9						
	Sesquiterpene hydrocarbons			80.9						
	Nor oxygenated sesquiterpenes			0.1						
	Oxygenated sesquiterpenes			8.7						
	Oxygenated Diterpenes hydrocarbons			0.7						
	Other			2.4						
	Total Identified			96.0	98.9	90.8	98.8	92.7	92.0	97.5

RI_exp_—retention indices experimentally obtained by C_8_–C_32_ *n-*alkanes series; RI_lit_—retention indices literally (RI library Adams4 and Nist webbook); * sum of compounds not detected in sample from this study.

**Table 6 molecules-28-04611-t006:** Traditional use of *Teucrium chamaedrys* and scientifically proven activities.

Traditional Use			Scientifically Proven Activities		
Country/Ailments Treated	Part Used/Preparation/Administration	Reference	Activity	Form	Reference
Turkey: toothache, kidney pain, stomachache, indigestion, hemorrhoids, heart diseases	herb (aerial parts)/infusion, decoction, fresh/internally	[30,31]	Thyrosinase inhibitory effect	ethanol extract	[63]
Bosnia and Herzegovina: digestive aliments: spasm relief, liver and gall ailments, diarrhea, heartburn, dry cough, influenza infections	aerial parts/infusion, fresh juice/internally	[10,57,58,59,60]	Moderate cytotoxic activity against cervix carcinoma, rhabdomyosarcoma and murine fibroblast cells	ethanol extract	[63]
Serbia: digestive complains: liver, spleen and gall complaints, diarrhea, loss of appetite, stomachache, hemorrhoids, against ulcers, respiratory aliments bronchitis, tuberculosis, fever, vaginal infections, chronic inflammation of the mucous membranes in the eyes and nose, against gout, as antitoxin (against snake bite)	aerial parts, leaf/infusion/internally; externally an astringent infusion	[4,5,6,11,61,62,74,75]	Antioxidant	water, methanol, ethyl-acetate, acetone and petroleum ether extracts	[76]
Kosovo: digestive complains: stomachache, antidiarrheal, antihemorrhoids, antidiabetic, appetizing, respiratory inflammation	flowering aerial parts, leaves/infusion/internally	[55,64,77,78,79]	Antimicrobial	essential oil	[80]

**Table 7 molecules-28-04611-t007:** Volatile compounds in *Marrubium peregrinum* from Rtanj Mt, Serbia (this study—TS) and references data.

No	Chemical Compound	RI_exp_	RI_lit_	Rtanj, Serbia (TS)	Bačko Gradište, Serbia [82]	Novi Kneževac, Serbia [82]	Senta, Serbia [82]	Domokos, Greece [81]	Parnassos, Greece [81]	Slovakia [83]
1	*α*-Pinene	931	932	0.5	0.4	0.3	0.3	-	-	-
2	Camphene	945	946	0.1	0.1	0.1	0.1	-	-	-
3	Sabinene	970	969	0.1	-	0.1	0.1	-	-	-
4	*β*-Pinene	973	974	0.6	0.5	0.5	0.5	-	-	-
5	2-Pentyl furan	989	984	0.1	-	-	-	-	-	-
6	3-Octanol	991	988	0.2	-	-	-	-	-	-
7	*α*-Terpinene	1014	1014	0.1	-	-	-	-	-	-
8	*p*-Cymene	1021	1020	0.2	-	-	-	-	-	-
9	1,8-Cineole	1027	1026	1.3	-	-	-	-	-	-
10	*γ*-Terpinene	1054	1054	0.1	-	-	-	-	-	-
11	Artemisia ketone	1056	1056	0.4	-	-	-	-	-	-
12	Linalool	1097	1095	0.3	0.2	0.3	0.3	1.5	1.7	1.4
13	2-Methyl butyl-2-methyl butyrate	1101	1100	0.1	-	-	-	-	-	-
14	*cis*-Thujone	1103	1101	0.4	1.5	1.3	1.7	-	-	-
15	*trans*-Thujone	1114	1112	25.1	2.1	2.3	3.3	-	-	-
16	*α*-Camphoenal	1122	1122	0.1	-	-	-	-	-	-
17	*iso*-3-Thujanol	1129	1134	0.2	-	-	-	-	-	-
18	*trans*-Pinocarveol	1133	1135	0.3	-	-	-	-	-	-
19	Camphor	1139	1141	0.7	-	-	-	-	-	-
20	Pinocarvone	1156	1158	0.4	-	-	-	-	-	-
21	Borneol	1159	1165	0.7	-	-	-	-	-	-
22	Terpinen-4-ol	1171	1174	0.1	-	-	-	-	-	-
23	*α*-Terpineol	1184	1186	0.1	-	-	-	0.1	0.2	-
24	Myrtenal	1190	1195	0.2	-	-	-	-	-	-
25	*cis*-Chrysanthenyl acetate	1255	1261	0.1	-	-	-	-	-	-
26	Geranial	1265	1264	0.1	-	-	-	-	-	-
27	Bornyl acetate	1279	1287	0.6	-	-	-	-	-	-
28	Thymol	1285	1289	1.0	-	-	-	-	-	-
29	Carvacrol	1295	1298	0.5	1.3	1.4	1.6	-	-	-
30	*δ*-Elemene	1331	1335	0.7	-	-	-	-	-	-
31	*α*-Copaene	1369	1374	0.3	0.3	0.3	0.2	0.3	0.4	0.4
32	*β*-Elemene	1386	1389	0.2	-	-	-	0.6	1.3	-
33	Methyl eugenol	1399	1403	0.1	-	-	-	-	-	-
34	*cis*-Caryophyllene	1400	1406	0.1	-	-	-	-	-	-
35	*trans*-Caryophyllene	1415	1417	32.4	13.2	14.3	18.0	0.6	0.7	31.3
36	*α*-Humulene	1447	1452	2.6	2.0	1.9	2.6	-	-	2.4
37	*trans*-*β*-Farnesene	1451	1454	0.5	3.7	4.4	5.1	24.2	21.5	-
38	Germacrene D	1476	1484	0.6	6.8	8.6	9.1	-	4.8	28.1
39	NI-1	1477	/	1.0	-	-	-	-	-	-
40	*β*-Selinene	1481	1489	0.6	-	-	-	-	-	-
41	Bicyclogermacrene	1491	1500	5.0	7.6	6.4	9.8	11.0	4.8	15.3
42	*β*-Bisabolene	1503	1505	0.1	-	-	-	1.4	1.1	-
43	Davana ether	1507	1517	0.3	-	-	-	-	-	-
44	*δ*-Cadinene	1518	1522	0.2	1.3	1.6	1.7	1.8	1.4	1.4
45	NI-2	1537	/	3.9	-	-	-	-	-	-
46	Elemol	1543	1548	0.2	-	-	-	-	-	-
47	*7-epi-trans*-Sesquisabinene hydrate	1547	/	0.3	-	-	-	-	-	-
48	1,5-Epoxysalvial-4(14)-ene	1561	1561	0.1	-	-	-	-	-	-
49	NI-3	1571	/	3.4	-	-	-	-	-	-
50	Caryophyllene oxide	1576	1582	2.4	4.2	3.7	5.0	-	-	2.8
51	Davanone	1580	1587	0.1	-	-	-	-	-	-
52	Salvial-4(14)-en-1-one	1587	1594	0.2	-	-	-	-	-	-
53	NI-4	1612	/	1.7	-	-	-	-	-	-
54	Muurola-4,10(14)-dien-1-*β*-ol	1620	1630	0.2	-	-	-	-	-	-
55	*cis*-Cadin-4-en-7-ol	1627	1635	0.4	-	-	-	-	-	-
56	6-Methyl-6-(3-methylphenyl)-heptan-2-one	1631	1639	0.2	-	-	-	-	-	-
57	*epi*-*α*-Cadinol (=*τ*-Cadinol)	1634	1638	0.1	-	-	-	-	-	-
58	*α*-Bisabolone oxide A	1677	1684	0.5	-	-	-	-	-	-
59	Eudesma-4(15),7-dien-1*β*-ol	1680	1687	0.5	-	-	-	-	-	-
60	Davanol acetate	1685	1689	0.2	-	-	-	-	-	-
61	6R,7R-Bisabolone	1737	1740	0.1	-	-	-	-	-	-
62	6S,7R-Bisabolone	1741	1448	0.3	-	-	-	-	-	-
63	6,10,14-trimethyl-2-Pentadecanone	1841	1847	0.4	-	-	-	-	-	-
64	5E,9E-Farnesyl acetone	1917	1913	0.1	-	-	-	-	-	-
	Other *			-	38.5	40.1	36.8	45.4	43.3	15.0
	Monoterpene hydrocarbons			1.7						
	Oxygenated monoterpenes			32.6						
	Sesquiterpene hydrocarbons			43.3						
	Phenylpropanoids			0.1						
	Oxygenated sesquiterpenes			6.0						
	Other			1.0						
	Total Identified			94.7	83.7	87.6	96.2	86.9	81.2	98.1

RI_exp_—retention indices experimentally obtained by C_8_–C_32_ *n-*alkanes series; RI_lit_—retention indices literally (RI library Adams4 and Nist webbook); * sum of compounds not detected in sample from this study. NI-1: 189(100), 133(63), 204(45), 91(33), 93(32), 107(28), 147(27), 105(26), 109(19), 79(19). NI-2: 107(100), 132(53), 91(34), 105(33), 119(29), 133(28), 125(28), 122(25), 93(21), 41(20). NI-3: 119(100), 91(94), 105(80), 107(80), 93(70), 132(66), 159(64), 43(61), 131(57), 41(56). NI-4: 124(100), 109(98), 81(88), 95(74), 82(61), 107(53), 67(52), 41(50), 93(44), 55(40).

**Table 8 molecules-28-04611-t008:** Traditional use of *Marrubium peregrinum* and scientifically proven activities.

Traditional Use			Scientifically Proven Activities		
Country/Ailments Treated	Part Used/Preparation/Administration	Reference	Activity	Form	Reference
Serbia: menstrual difficulties (bladder or uteral pain), anemia, against hemorrhoids, digestive complaints, tonic, excitant, resolvent, secretory stimulant, respiratory tract (catarrh, cough), arrhythmia, overall weakness	herb/infusion/internally	[7,74,85]	Antioxidant	water, methanol, ethyl-acetate, acetone and petroleum ether extracts, essential oil	[82,88]
Bulgaria: medicinal and spice plant, for garden brooms	ns; technical plant	[86,87]	Antimicrobial	acetone, ethyl acetate and methanol extracts	[89]

ns—not specified.

## Data Availability

Not applicable.

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
