# Peer review of "Screening of Volatile Compounds, Traditional and Modern Phytotherapy Approaches of Selected Non-Aromatic Medicinal Plants (Lamiaceae, Lamioideae) from Rtanj Mountain, Eastern Serbia"

_molecules, 2023, doi:10.3390/molecules28124611_

Round 1

Reviewer 1 Report

Screening of volatile compounds, traditional and modern phytotherapy approach of selected non-aromatic medicinal plants (Lamiaceae, Lamioideae) from Rtanj Mountain, Eastern Serbia

It is a relevant study, however, some corrections are needed:

Abstract needs to be improved, insert the expressive results achieved in this research, as well as methodology employed;

In line 66-68 further specify the use of this species in the treatment of diseases, for example, for anemia it is not clear whether it is used in the form of tea or other preparations;

On line 73. What would these traditional medicines be?

Table 1.  It is necessary to insert the RI of the literature and to determine the total of the chemical classes (monoterpenes  and sesquiterpenes);

Figure 1.  Improve sharpness (resolution of images);

Table 3. It is necessary to insert the RI of the literature and to determine the total of the chemical classes (monoterpenes and sesquiterpenes);

Table 5.  It is necessary to insert the RI of the literature and to determine the total of the chemical classes (monoterpenes  and sesquiterpenes);

Table 7.  It is necessary to insert the RI of the literature and to determine the total of the chemical classes (monoterpenes  and sesquiterpenes);

The conclusion needs to be improved as it looks like an introduction. Enter the main and expressive results of this search.

Author Response

Thank you very much for your valuable comments and feedback regarding our research paper. Each of your insights have served to strengthen our manuscript and we have made changes to reflect them. We have made the changes directly to the manuscript, as well as recorded the changes below, please see the attachment.

Reviewer 2 Report

The manuscript "Screening of volatile compounds, traditional and modern phytotherapy approach of selected non-aromatic medicinal plants (Lamiaceae, Lamioideae) from Rtanj Mountain, Eastern Serbia" is devoted to study of chemical composition of four medicinal plants: Ironwort (Sideritis montana L.), mountain germander (Teucrium montanum L.), wall germander (Teucrium chamaedrys L.) and horehound (Marrubium peregrinum L.) growing in Serbia. The work presents data on the components of essential oil identified by GC-MS and GC-FID. In general, the work is an enumeration of the detected components in different plant species. Despite the relatively large amount of information, the scientific significance is not too great. If data on the components of different plant species are presented in this work, a more detailed analysis of the differences should be made, the reasons for such differences should be considered in more depth, or experimental data on the biological activity of the essential oils of the studied plants should be added.

It should be noted that the manuscript is written in a good language, has a good structure and a clear representation of the data. I think, this manuscript can be published in the Molecules journal after major revision after taking into account general recommendations and comments given below:

1.      Figure 1: This figure has a low informative value. It's not entirely clear what the authors want to show on it. If we are talking about the difference in compositions, then this can be seen from the tables.

2.      The percentage of convergence in the definition of components must be specified.

3.      Why do the authors choose the method of drying plants indicated in the work (dried hung upside down in a shaded, well-ventilated place for one week)? If the purpose of the work is to identify volatile components, collection and sample preparation should be carried out in a different way.

Author Response

(The authors gave the same response as above.)

Round 2

Reviewer 1 Report

Improvements were made to the manuscript. However, the conclusion remains extensive, it is necessary for the authors to point out the main and significant results of the research

Reviewer 2 Report

I thank the authors for the work done.